# Interpersonal psychotherapy delivered by nonspecialists for depression and posttraumatic stress disorder among Kenyan HIV–positive women affected by gender-based violence: Randomized controlled trial

**Susan M. Meffert**[1]*, **Thomas C. Neylan**[1,2], **Charles E. McCulloch**[3], **Kelly Blum**[1], **Craig R. Cohen**[4], **Elizabeth A. Bukusi**[5,6,7], **Helen Verdeli**[8], **John C. Markowitz**[9], **James G. Kahn**[10], **David Bukusi**[11], **Harsha Thirumurthy**[12], **Grace Rota**[13], **Ray Rota**[13], **Grace Oketch**[13], **Elizabeth Opiyo**[13], **Linnet Ongeri**[14]

1 Department of Psychiatry, University of California, San Francisco (UCSF), San Francisco, California, United States of America, 2 Mental Health Service, San Francisco Veterans Affairs Medical Center, San Francisco, California, United States of America, 3 Department of Epidemiology & Biostatistics, UCSF, San Francisco, California, United States of America, 4 Department of Obstetrics, Gynecology & Reproductive Sciences, Bixby Center for Global Reproductive Health, UCSF, San Francisco, California, United States of America, 5 Departments of Obstetrics and Gynecology, University of Washington, Seattle, Washington, United States of America, 6 Center for Microbiology Research, Kenya Medical Research Institute (KEMRI), Nairobi, Kenya, 7 Department of Obstetrics and Gynecology, Aga Khan University, Nairobi, Kenya, 8 Teachers College, Columbia University, New York, New York, United States of America, 9 Department of Psychiatry, Vagelos College of Physicians and Surgeons, Columbia University, New York, New York, United States of America, 10 Philip R. Lee Institute for Health Policy Studies, Global Health Sciences, and Global Health Economics Consortium, UCSF, San Francisco, California, United States of America, 11 Department of Psychiatry, Kenyatta National Hospital, University of Nairobi, Nairobi, Kenya, 12 Perelman School of Medicine, University of Pennsylvania, Philadelphia, Pennsylvania, United States of America, 13 University of Nairobi, Nairobi, Kenya, 14 KEMRI, Nairobi, Kenya

* Susan.Meffert@ucsf.edu

## Abstract

### Background

HIV–positive women suffer a high burden of mental disorders due in part to gender-based violence (GBV). Comorbid depression and posttraumatic stress disorder (PTSD) are typical psychiatric consequences of GBV. Despite attention to the HIV-GBV syndemic, few HIV clinics offer formal mental healthcare. This problem is acute in sub-Saharan Africa, where the world's majority of HIV–positive women live and prevalence of GBV is high.

### Methods and findings

We conducted a randomized controlled trial at an HIV clinic in Kisumu, Kenya. GBV-affected HIV–positive women with both major depressive disorder (MDD) and PTSD were randomized to 12 sessions of interpersonal psychotherapy (IPT) plus treatment as usual (TAU) or Wait List+TAU. Nonspecialists were trained to deliver IPT inside the clinic. After 3 months, participants were reassessed, and those assigned to Wait List+TAU were given IPT. The

**Data Availability Statement:** Study data cannot be shared publicly because of Institutional Review Board (IRB) restrictions. Data are available from the UCSF IRB for researchers who meet criteria for access to confidential data. The data underlying the results presented in the study are available from Ms. Rachel Burger: rachel.burger@ucsf.edu.

**Funding:** This work was supported by two awards to SMM: (1) National Institutes of Mental Health-K23MH098767 and (2) Building Interdisciplinary Research Careers in Women's Health (BIRCWH-UCSF) Award. The funders had no role in study design, data collection and analysis, decision to publish, or preparation of the manuscript The funders had no role in study design, data collection and analysis, decision to publish, or preparation of the manuscript.

**Competing interests:** The authors have declared that no competing interests exist.

**Abbreviations:** AUDIT, Alcohol Use Disorders Identification Test; BDI-II, Beck Depression Index; CBT, cognitive behavioral therapy; CTS2, Conflict Tactics Scale; DAST, Drug Abuse Screening Test; FACES, Family AIDS Care, Education, and Services; GBV, gender-based violence; ICF, International Classification of Functioning, Disability, and Health; IPT, interpersonal psychotherapy; IPV, intimate partner violence; LMIC, low- and middle-income countries; LTFU, lost to follow-up; MDD, major depressive disorder; OR, odds ratio; PCL-C, Posttraumatic Stress Disorder Checklist-Civilian; PEPFAR, President's Emergency Plan for AIDS Relief; PLWH, people living with HIV; PTSD, posttraumatic stress disorder; SAS, Social Adjustment Scale; SSA, sub-Saharan Africa; TAU, treatment as usual; THQ, Trauma History Questionnaire; WHODAS, World Health Organization Disability Assessment Schedule.

primary outcomes were diagnosis of MDD and PTSD (Mini International Neuropsychiatric Interview) at 3 months. Secondary outcomes included symptom measures of depression and PTSD, intimate partner violence (IPV), and disability. A total of 256 participants enrolled between May 2015 and July 2016. At baseline, the mean age of the women in this study was 37 years; 61% reported physical IPV in the past week; 91% reported 2 or more lifetime traumatic events and monthly income was 18USD. Multilevel mixed-effects logistic regression showed that participants randomized to IPT+TAU had lower odds of MDD (odds ratio [OR] 0.26, 95% CI [0.11 to 0.60], $p = 0.002$) and lower odds of PTSD (OR 0.35, [0.14 to 0.86], $p = 0.02$) than controls. IPT+TAU participants had lower odds of MDD-PTSD comorbidity than controls (OR 0.36, 95% CI [0.15 to 0.90], $p = 0.03$). Linear mixed models were used to assess secondary outcomes: IPT+TAU participants had reduced disability (−6.9 [−12.2, −1.5], $p = 0.01$), and nonsignificantly reduced work absenteeism (−3.35 [−6.83, 0.14], $p = 0.06$); partnered IPT+TAU participants had a reduction of IPV (−2.79 [−5.42, −0.16], $p = 0.04$). Gains were maintained across 6-month follow-up. Treatment group differences were observed only at month 3, the time point at which the groups differed in IPT status (before cross over). Study limitations included 35% attrition inclusive of follow-up assessments, generalizability to populations not in HIV care, and data not collected on TAU resources accessed.

## Conclusions

IPT for MDD and PTSD delivered by nonspecialists in the context of HIV care yielded significant improvements in HIV–positive women's mental health, functioning, and GBV (IPV) exposure, compared to controls.

## Trial registration

Clinical Trials Identifier NCT02320799.

## Author summary

### Why was the study done?

- HIV–positive women in sub-Saharan Africa experience high levels of gender-based violence (GBV), leading to a very high prevalence of mental disorders, particularly depression and posttraumatic stress disorder (PTSD).

- Despite knowledge that evidence-based psychotherapy for depression and PTSD can be delivered by local nonspecialists in East Africa with high efficacy, little data exists on scalable treatment models for HIV–positive women affected by GBV in the region.

### What did the researchers do and find?

- We partnered with a large HIV clinic in western Kenya to conduct a randomized controlled trial of interpersonal psychotherapy (IPT) versus Wait List-treatment as usual

(TAU). Participants were 256 women enrolled in HIV care and affected by GBV who met criteria for major depressive disorder (MDD) and PTSD (primary outcomes).

- We used a scalable intervention in which local nonspecialists (no prior mental health training required) were trained to deliver IPT inside an HIV clinic, working closely with HIV clinic staff and providers.

- At the conclusion of treatment, those who received IPT had significant reduction in MDD, PTSD, and combined MDD-PTSD compared to Wait List+TAU controls. Wait List+TAU participants experienced similar remission after they received IPT treatment and gains were maintained across follow-up.

- Secondary findings: Compared with controls, IPT participants had a greater reduction of disability, intimate partner violence, and nonsignificantly reduced work absenteeism.

**What do these findings mean?**

- This study suggests that IPT can be delivered in a scalable manner, including administration by nonspecialists, housed within existing HIV clinics.

- Delivering IPT to HIV–positive women affected by GBV using clinic-integrated non-specialists can achieve substantial remission of MDD and PTSD sustained over 6-month follow-up, with apparent reductions in disability and physical violence by intimate partners.

## Introduction

### Background

People living with HIV (PLWH) suffer from mood and anxiety disorders at 3 to 5 times the prevalence of general populations [1,2]. Among PLWH, HIV–positive women suffer even higher rates of mental disorders partly because of high rates of intimate partner violence (IPV) against HIV–positive women [3]. Mental disorders affect many gender-based violence (GBV) survivors, typically manifesting as depression and posttraumatic stress disorder (PTSD) [4]. Despite attention to the HIV-GBV syndemic and awareness that depression and PTSD significantly worsen HIV outcomes [5,6], few HIV clinics have integrated formal mental healthcare.

This problem is particularly acute in sub-Saharan Africa (SSA), where most of the world's HIV–positive women live [7] and where GBV prevalence is highest [8]. Many HIV–positive women develop depression and PTSD—intensifying the public health urgency is the scarcity of public sector mental healthcare [9,10]. Even in general populations of PLWH in SSA, depression and PTSD prevalence are high: A meta-analytic study ($n$ = 60,476) of depression among PLWH in SSA found a 36% prevalence [2]. A study among PLWH in Zimbabwe found that more than half met criteria for probable PTSD [11]. Compounding the problem, low- and middle-income countries (LMICs), including those in SSA, have a 75% treatment gap—75% of individuals in LMICs with common mental disorders, such as depression or PTSD, never receive treatment at all [12]. Indeed, LMICs carry the great preponderance of world mental health disability due to the coupling of high prevalence with this treatment gap.

More than a decade of research shows that local nonspecialists can provide efficacious, evidence-based psychotherapy for depression and anxiety in LMICs [13]. More recently, studies in SSA have demonstrated that non-mental health specialists can deliver interventions to decrease common mental disorder symptoms in primary care clinics with high HIV prevalence and improve adherence among PLWH [14–16]. A recent cluster randomized trial ($n$ = 1,473) of nonspecialist group support psychotherapy for major depressive disorder (MDD) among PLWH in Uganda found it more effective than group HIV education [17]. Yet, to our knowledge, no integrated treatment studies exist for formally diagnosed comorbid MDD and PTSD—the most common mental disorders for traumatized individuals—and extremely prevalent among the many HIV+GBV+ women living in SSA.

Our study site was the Family AIDS Care, Education, and Services (FACES) HIV care and clinical research clinic in Kisumu, Kenya. Funded by the President's Emergency Plan for AIDS Relief, FACES is a 12-year collaboration between the University of California San Francisco and the Kenya Medical Research Institute serving more than 140,000 HIV–positive individuals in the Nyanza region of Kenya. This region has the country's highest adult HIV prevalence, reaching 21% in some areas, and the second highest prevalence of physical and/or sexual violence against women by an intimate partner—51.9% of women aged 15 to 49.

## Mental healthcare needs study

In 2013, we conducted a qualitative mental healthcare needs assessment of HIV+GBV + women served by the large, Kisumu FACES clinic, using focus groups and key informant interviews [18]. Respondents included HIV–positive female clinic attendees, clinic HIV care providers, medication adherence counselors, hospital leadership, and community leaders. Most interviewees reported that physical, emotional, and sexual violence against HIV–positive women were common in the region and named depression, anxiety, traumatic stress symptoms, and suicidal thoughts as resulting mental health problems among HIV+GBV+ women. Respondents believed that HIV+GBV+ women needed mental health treatment and the great majority preferred weekly individual counseling at the HIV clinic as a treatment modality. Seeking social support was the dominant method of coping with emotional distress. As previously reported [18,19], these data informed our selection and adaption of interpersonal psychotherapy (IPT) delivered by local non-mental health specialists within the HIV clinic to treat MDD and PTSD symptoms of female HIV+GBV+ clinic attendees.

Delivering mental healthcare for diagnosed disorders inside the HIV clinic was a clear preference reported in our needs assessment study and has a strong evidence base [18,20]. Locating mental healthcare in an HIV clinic has additional advantages in the context of East African culture and structure of HIV clinic operations: Attendees typically consider HIV clinics "safe" locations to receive care for stigmatized diseases. Because both mental disorders and HIV infection are highly stigmatized in SSA, combining care at an HIV clinic may help patients feel safer and more willing to attend treatment. Unlike much regional primary healthcare focusing on one-time visits for acute conditions, East African HIV clinics typically provide patients' medication monthly, meaning patients return to the clinic regularly when not acutely ill to obtain medication and see HIV clinicians. This characteristic of the HIV clinics is syntonic with mental healthcare—treatment for MDD and PTSD is longitudinal and requires multiple visits when physical illness is non-acute.

## Comparator condition

Treatment As Usual (TAU) comparators for psychotherapy trials are the subject of controversy [21,22]. In choosing a Wait List+TAU control condition for this study, we note that

recent psychotherapy trial guidelines advise against creating "de novo" control interventions that are not genuine local clinical alternatives, because the results of such studies would not inform care for target populations [21]. For studies that examine new effects of an intervention such as IPT for comorbid MDD and PTSD among HIV+GBV+ women, current guidance indicates that stakeholder interests are best served by allowing the intervention to demonstrate its effects without an overly formidable comparator that could prevent further investigation of a beneficial treatment [21,22]. Avoiding this potential harm to stakeholders is particularly important when working in a low-resource setting such as western Kenya where public sector treatment access is scarce. We note that selection of Wait List+TAU as our comparator condition does lead to other risks, such as lack of control over the treatment received in the Wait List+TAU group and related threats to internal validity [22]. In this case, we concluded that these and other sacrifices were worth the potential stakeholder gain achieved through the development of useful treatment for the most common mental disorders among HIV+GBV + women in Kenya.

## Objective

The study aimed to test a sustainable, scalable model for treating comorbid MDD and PTSD among GBV-affected HIV–positive women. We conducted a randomized, effectiveness-implementation type I trial of IPT plus TAU delivered by nonspecialists in an HIV clinic compared to Wait List+TAU for care-engaged HIV–positive women (Fig 1 and study design, below; see S1 Table). Effectiveness-implementation hybrid studies have developed over the past decade to collect both clinical and implementation data simultaneously. The goal of these designs is to speed translation of positive findings toward scale up and benefit to public health by collecting implementation data earlier in the pipeline than traditionally done, allowing for faster recognition of scalable (and not scalable) interventions [19,23]. "Type I" effectiveness-implementation hybrid designs (as with this study) emphasize effectiveness outcomes, while collecting initial data on implementation parameters [23].

After completing baseline assessments on all participants, IPT was offered to the IPT+TAU group. After the 3-month, posttreatment assessment of those initially assigned to IPT, the Wait List+TAU group was offered IPT. We hypothesized IPT+TAU would be more effective than Wait List+TAU for remission of MDD and PTSD at month 3. Assessments were repeated at months 6 and 9 to evaluate IPT effect on the Wait List+TAU group and maintenance of gains for the IPT+TAU group. Given effectiveness-implementation design, our primary outcome of interest was remission (absence of disease)—the usual goal of "real-life" psychiatric practice.

## Methods

### Study design

As detailed in the published study protocol [24], the effectiveness aspect of the study included a mental health clinical core, neurocognitive testing, HIV treatment adherence, and HIV viral load on a participant subsample, while the implementation domains were assessed using qualitative analyses of acceptability and appropriateness, as well as quantitative measures of feasibility, treatment fidelity and economic productivity, and treatment cost-benefit analyses. Given the breadth of outcome data, we are not able to report all the study's findings with adequate discussion here. Therefore, we focus on the mental health clinical core, which included the study's primary outcomes and key secondary outcomes, as well as implementation outcomes with direct relevance to the intervention—feasibility and fidelity.

**Table 1. Outcomes.**

| | Prespecified outcomes | Reported in this manuscript |
|---|---|---|
| **Demographics** | Age, household monthly income | X Note: Added question on payment of school fees (indicator of economic health) |
| **Effectiveness Domains** | | |
| Mental Health Clinical Core | AUDIT, BDI, CTS, DAST, MINI, PCL, THS, WHOQoL-BREF, WHODAS | X Note: WHOQoL-BREF was dropped due to redundancy and participant burden |
| Neurocognitive Battery | Category fluency, Color Trails, Grooved Pegboard, WAIS-III, Digit Span, WHO/UCLA AVLT, BVMT, WHO/UCLA AVLT Delayed Recall | |
| HIV Outcomes | ART treatment adherence, HIV viral load (participant subset) | |
| **Implementation Outcomes** | | |
| Acceptability and Appropriateness | Qualitative interviews and focus groups* | |
| Feasibility | Pre-specified benchmarks* | X |
| Fidelity | IPT adherence monitoring of nonspecialist delivery using individual and group formats* | X |

AUDIT, Alcohol Use Disorders Identification Test; ART, Antiretroviral Therapy; BDI, Beck Depression Index; BVMT, Brief Visuospatial Memory Test; CTS, Conflict Tactics Scale; Digit Span; DAST, Drug Abuse Screening Test; HIV, Human Immunodeficiency Virus; MINI, Mini International Neuropsychiatric Interview; PCL, Posttraumatic Stress Disorder Checklist; THQ, Trauma History Questionnaire; THS, Trauma History Screen; WAIS-III, Wechsler Adult Intelligence Scale; WHODAS, World Health Organization Disability Adjustment Scale; WHOQoLBREF, World Health Organization Quality of Life Brief; WHO/UCLA AVLT, World Health Organization-University of California Los Angeles Auditory Verbal Learning Test; WHO/UCLA AVLT Delayed Recall, World Health Organization-University of California Los Angeles Auditory Verbal Learning Test-Delayed Recall.

*See details in published protocol (reference #22).

Participants randomized to IPT+TAU were allocated to 12 weekly IPT sessions provided by nonspecialists (description of nonspecialists, below) trained to deliver IPT in the HIV clinic. Nonspecialist delivery of psychotherapy is a common strategy for testing models of scalable mental healthcare in low-resource settings with few mental health professionals [13,25,26]. The control group received Wait List+TAU, then received IPT following 3-month assessment. All study participants received TAU throughout the study (Fig 1).

## Ethics

The UCSF Committee on Human Research (IRB# 13–11765) approved the protocol on March 17, 2014) and the Kenya Medical Research Institute (KEMRI) Scientific Ethical Review Unit (SSC Protocol No. 2753) approved the protocol on May 26, 2014. Secondary to literacy rates in the region, written or thumbprint informed consent was used for all participants. Prospective participants were also assessed for their understanding of each point of the informed consent: they were asked to repeat each point of the informed consent in their own words (three efforts allowed) and the point was initialed by the research staff.

## Participants

**Inclusion criteria.** Participants were HIV–positive women receiving FACES-Kisumu HIV care, at least 18 years old, diagnosed with MDD and PTSD in the context of GBV [27] on the Mini International Neuropsychiatric Interview (MINI 5.0), who were able to attend 12 once weekly therapy sessions and provide written informed consent.

**Exclusion criteria.** Cognitive dysfunction requiring a higher level of care; severe thought or mood disorder symptoms requiring higher level of care or interfering with IPT

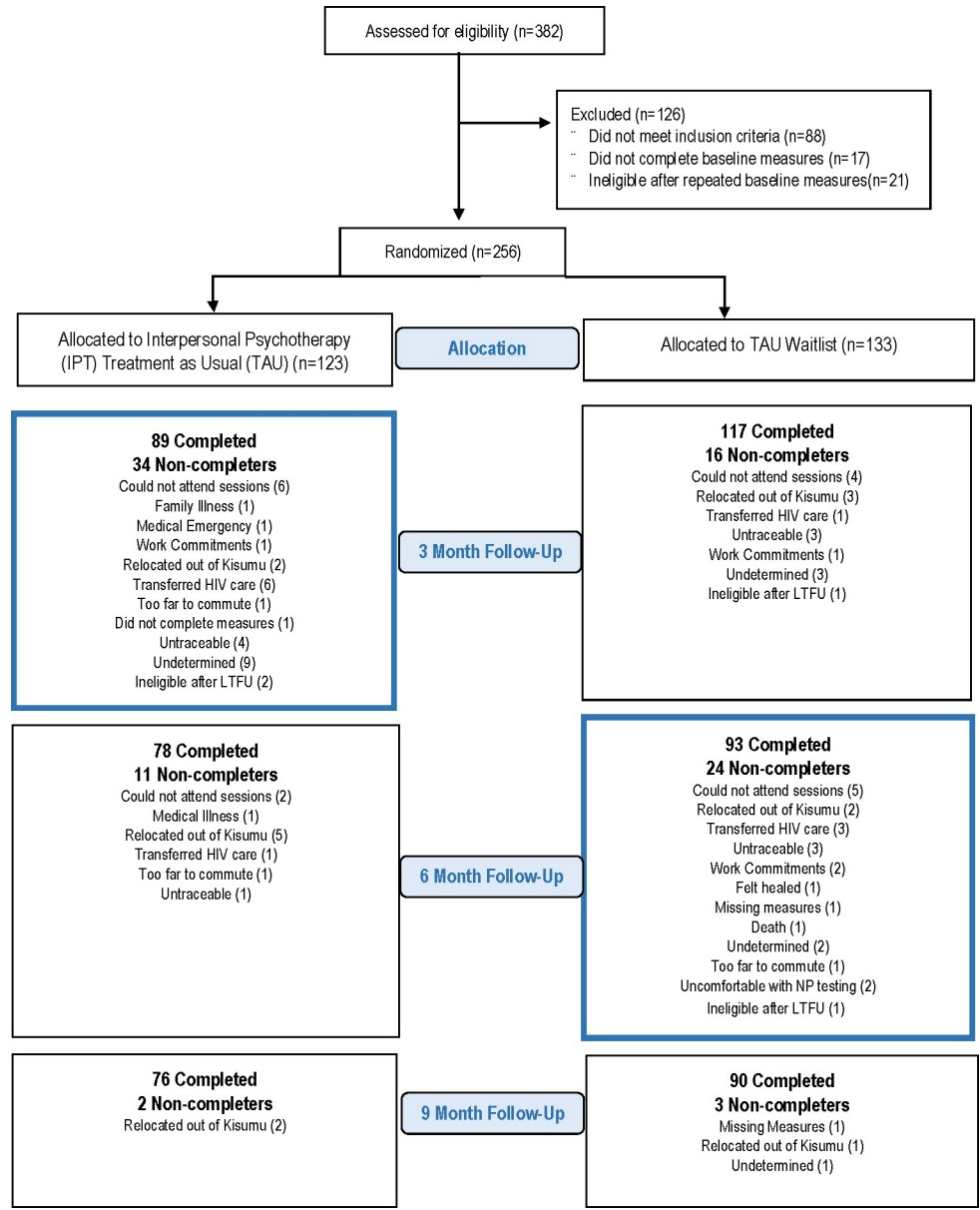

**Fig 1. Participant flow.**

participation; and current drug and alcohol dependence requiring substance use treatment. Cognitive dysfunction and severe thought disorder were screened by clinical evaluators using orientation questions. Participants were asked to state their name, location and the approximate date, as well as their reason for meeting with the team. If they could not answer these questions, they were referred for medical evaluation. Current alcohol and drug dependence were assessed by the clinical evaluators using the Alcohol Use Disorders Identification Test (AUDIT) and the Drug Abuse Screening Test (DAST), with cutoffs of 8 and 3, respectively, indicating harmful/hazardous alcohol use/moderate drug use (Table 2). All participants were screened for suicidality and other acute mental conditions, as well as any immediate risk of harm from GBV. If they screened positive, they were transported using study funds and with

**Table 2. Baseline characteristics.**

| Baseline Characteristics (*n* = 256, total) | IPT+TAU (*n* = 123) | | Wait List+TAU (*n* = 133) | |
|---|---|---|---|---|
| | Mean or No. (SD or %) | Non-missing/Total | Mean or No. (SD or %) | Non-missing/Total |
| Average age in years | 36.9 (9.4) | 120/123 | 37.0 (9.4) | 131/133 |
| Depression symptoms (BDI II) | 27.5 (10.5) | 121/123 | 29.0 (9.6) | 132/133 |
| PTSD symptoms (PCL-C) | 56.6 (15.5) | 122/123 | 56.1 (16.8) | 132/133 |
| Physical intimate partner violence in past week among partnered participants (CTS)* | 37 (66.1%) | 120/123 | 27 (55.1%) | 132/133 |
| Physical violence (partner to participant) among partnered participants** | 34 (60.7%) | 120/123 | 27 (55.1%) | 132/133 |
| Physical violence (participant to partner) among partnered participants** | 18 (32.1%) | 120/123 | 14 (28.6%) | 132/133 |
| Two or more different types of lifetime trauma, e.g., crime, sexual/physical assault, disaster (THQ) | 113 (91.9%) | 123/123 | 120 (90.2%) | 133/133 |
| At least 4 different types of lifetime trauma, e.g., crime, sexual/physical assault, disaster (THQ) | 40 (32.5%) | 123/123 | 48 (36.1%) | 133/133 |
| Alcohol use (AUDIT ≥8, harmful or hazardous drinking) | 0.63 (2.88) | 103/123 | 0.46 (1.89) | 119/133 |
| Drug use (DAST ≥3, moderate and up) | 1.20 (0.88) | 54/123 | 1.43 (1.19) | 80/133 |
| Disability score (WHODAS) | 30.1(18.6) | 121/123 | 29.1(16.9) | 131/133 |
| <u>Economic productivity and absenteeism</u> | | | | |
| Monthly income (USD) | 18.1 (4.3) | 122/123 | 18.1 (4.1) | 133/133 |
| Days in the past month partially or completely unable to work | 8.3 (10.7) | 121/123 | 7.2 (9.3) | 131/133 |
| Paid school fees on time in past month for those with school age children | 17 (16.8%) | 101/123 | 19 (16.2%) | 117/133 |

AUDIT, Alcohol Use Disorders Identification Test; BDI II, Beck Depression Index; CTS, Conflict Tactics Scale; DAST, Drug Abuse Screening Test; PCL, PTSD Checklist; THQ, Trauma History Questionnaire; WHODAS, WHO Disability Assessment Schedule.

*Total number IPT+TAU partnered = 56, total number Wait List+TAU partnered = 49.

**Partner-to-participant and participant-to-partner violence can occur at the same time.

study staff escort to the local inpatient psychiatry unit, GBV evaluation or women's shelter for evaluation and treatment (S1 Text). Pregnant women were not excluded.

**Recruitment.** Patients were recruited through FACES HIV provider referral and self-referral following informational talks to FACES patients in the HIV clinic check-in and waiting areas. Consecutive HIV–positive women were consented and screened for inclusion.

**Compensation.** Many of the women in this study were economically disempowered and lived in rural areas, making weekly transport to the HIV clinic a significant financial challenge. Therefore, participants received 300ksh (3USD) to cover transport for their weekly IPT sessions and study assessments.

## Interventions

**Interpersonal psychotherapy (IPT).** IPT is a time-limited psychotherapy, often delivered as 12 weekly sessions, developed in the 1970s by Gerald Klerman and Myrna Weissman to treat MDD by addressing interpersonal difficulties and crises [28]. In Euro America, IPT is a first-line MDD treatment and showed better results for depressed HIV–positive patients than cognitive behavioral therapy (CBT) or supportive psychotherapy [29]. It demonstrated non-inferiority to exposure-based psychotherapy for PTSD, with advantages over prolonged exposure for patients with comorbid MDD [30]. IPT has a history of strong efficacy when delivered by nonspecialists in East African populations (e.g., [31]). IPT helps patients improve social functioning and manage interpersonal crises characterized as grief, role dispute, or role transition, by building social skills and gathering social support (see details in our prior publications)

[31]. The group randomized to IPT+TAU began weekly sessions after their baseline evaluation and continued until completing 12 IPT sessions. Participants could miss and make up 2 IPT sessions if they scheduled the make-up within 1 week of the missed session. Kenyan December holidays required 2 to 4 week breaks in IPT, depending on participants' travel. The 3-month assessment was completed at posttreatment, after 12 sessions of IPT were complete; the 24-week assessment occurred after the Wait List+TAU group completed IPT. Some posttreatment assessments occurred at a different time than others to adjust for missed and rescheduled sessions. IPT was conducted in private rooms adjacent to the HIV clinic. Sessions lasted approximately 60 minutes and the same therapist treated the participant throughout. Details of IPT used in the MIND study are described in a published case history from the study [32].

**Interpersonal psychotherapy (IPT) therapists.** Prospective IPT therapists were recruited with local advertisement by FACES. The local therapists trained to deliver IPT in this study needed only to have completed high school. All were women who underwent a 10-day training designed by study team IPT experts, followed by supervised practice cases. During the practice phase, each therapist received weekly telephone supervision from 1 of 2 psychiatrists expert in IPT (SMM and LO). Therapists were scored on 10 key aspects of delivering IPT, according to the IPT phase in which they were working, using a 10-point Likert scale. Only therapists consistently scoring 9 to 10 by the end of their IPT case progressed to work with enrolled study participants. Therapists not meeting this criterion treated an additional 1 to 2 pilot IPT cases and did not treat study participants unless they achieved competency (defined as scores of $\geq$9).

During the trial, IPT therapists continued weekly individual telephone supervision and IPT protocol adherence rating by the expert psychiatrists. IPT adherence was scored based on treatment phase (initial, formulation, middle, and termination). Adherence monitoring included assessment for use of "off-protocol" techniques and specifically evaluated for use of CBT and/or exposure therapy. A train-the-trainer model provided selected IPT therapists with training as on-site peer supervisors. These peer supervisors ran weekly, on-site IPT group supervision and adherence monitoring conducted as an adjunct to ongoing weekly telephone supervision.

**Treatment as usual (TAU).** At the FACES clinic, TAU resources for HIV+GBV+ women included informal counseling and HIV treatment adherence counseling, medical professionals, community elders, church leaders, police (e.g., GBV legal issues), and pro bono legal aid. These services and providers were available for access as needed. The TAU condition provided no explicit, evidence-based mental health treatment, although the social support implicit in each service is known to benefit mental health. While HIV clinicians can prescribe antidepressant medication, clinic attendees are rarely assessed or treated for mental disorders. After approximately 12 weeks of usual care, all Wait List+TAU participants who completed their second assessment then elected to receive IPT.

## Effectiveness outcomes

All participants in both study arms received identical assessments. Demographics (age, marital status, ethnic group, ART medications, other mental health treatment, and ART counseling received) and the trauma history questionnaire were administered at baseline; all other measures were assessed at baseline, 12 weeks, 24 weeks, and 36 weeks. The week 12 assessment took place after the IPT+TAU participants completed IPT and before the Wait List+TAU group started IPT. The week 24 assessment took place after the Wait List-TAU group had completed IPT. A team of study evaluators blinded to group assignment conducted all assessments. Given variable literacy in the region, all measures were translated into the local Dhuluo

and Kisawhili languages at the lowest expected adult education level and were read to participants. The study instruments have been validated in diverse populations. Additional psychometric testing and validation exceeded the study scope and budget.

## Primary outcomes (baseline and follow-up)

Mini International Diagnostic Interview (MINI 5.0) [33]. The MINI is a short, structured diagnostic interview developed in 1990 by psychiatrists and clinicians in the United States and Europe for DSM-IV and ICD-10 psychiatric disorders. The MINI 5.0 MDD and PTSD diagnostic modules were primary outcomes. We required meeting diagnostic criteria for both diagnoses for study inclusion.

## Secondary outcomes

Two continuous measures of depression and PTSD symptoms were used to check for convergent validity with primary outcomes: (1) Beck Depression Scale (BDI-II) [34] score. The BDI, developed in 1961, is a 21-item inventory assessing characteristic depressive attitudes and symptoms. (2) Posttraumatic Stress Disorder Checklist-Civilian (PCL-C) [35] score. The PCL is a 17-item measure of DSM-IV PTSD symptoms.

Conflict Tactics Scale (CTS2) [36]. The conflict tactics scale was created to measure negotiation and psychological and physical attacks by each partner in a marital, cohabiting, or dating relationship. We derived a measure of physical IPV from items 9 to 19 of the CTS (threat of or actual physical violence).

World Health Organization Disability Assessment Schedule (WHODAS) 2.0 [37]. The WHODAS 2.0 includes a 12-item instrument based on the International Classification of Functioning, Disability, and Health (ICF) that measures general health and disability levels, including mental and neurological disorders. Scoring involves summing the twelve 4-point Likert responses and dividing by 48, yielding a percentage that indicates level of disability, with 100% being maximal disability. The 2 final WHODAS 2.0 items record the number of days in the past month that the respondent was totally or partially unable to complete usual work activities, a measure of absenteeism.

## Baseline covariates

Drug and Alcohol Use. AUDIT [38]; DAST-10 [39].

Social Adjustment Scale (SAS) [40]. The SAS is a 54-item scale that assesses behavioral and emotional social adjustment across 6 major areas: work, leisure, extended family, primary relationship, and parental and family unit.

Trauma History Questionnaire (THQ) [41]. The THQ is a 24-item measure that evaluates respondents' experience of potentially traumatic events such as crime, general disaster, and sexual and physical assault using a yes/no format.

## Implementation outcomes

**Feasibility.** We prespecified feasibility benchmarks: 220 participants (original sample size) to be recruited by screening 300 participants; each completer attends 80% or more of IPT sessions, attrition is 50 participants or fewer, and one-half or fewer study therapists leave the study [24].

**Fidelity.** IPT adherence was scored using a 10-point Likert scale assessing adherence in each of the 3 IPT phases. Sessions were considered adherent to the IPT protocol if they averaged a score of 5 or higher and did not employ off-protocol interventions [24]. Sessions were

required to last 1 hour. Longer sessions were discouraged and shorter sessions were considered off-protocol. While audio recording of the sessions were made, due to unexpected expenses, the budget did not allow for a random 20% of sessions to be independently evaluated, as planned [24]. In place of this, we initiated local peer IPT supervision as an adjunct to and countercheck of adherence (see "Interventions" for details).

**Sample size.** In our prior work using a similar IPT treatment for a traumatized, low-resource population, we observed a Cohen's d effect size of 0.79 for depression symptom improvement [42]. While this effect size is large, it is consistent with other studies of evidence-based psychotherapy in LMICs [43–45]. The depression Cohen's d indicated need for a sample size of ≥50 for a 2-group design with power of 0.8 and probability level of 0.05. Recognizing the migration patterns of the study population and the study requirement to complete 4 assessments across 36 weeks, with substantial time investment at each assessment, we allowed for approximately 20% attrition at each assessment point and selected a sample size of 100. Because we collected venous blood samples for HIV viral load testing (data in preparation) in a subgroup of participants, which required multiple venipunctures, and fear of venipuncture can raise attrition, we increased our sample size by 50% per assessment point, estimating the need to screen approximately 380 participants to reach our recruitment goal.

**Randomization.** The study coordinator generated a random sequence of 2 colors in Excel corresponding to IPT+TAU and Wait List+TAU in blocks of 10. After completion of baseline measures, the study coordinator assigned participants to the next listed random color block and its corresponding treatment assignment. While the next assignment was visible to the study coordinator, no skipping was allowed and order of participant arrival was not altered.

**Blinding.** Staff who gathered baseline and subsequent assessment data were blinded to group assignment. The study coordinator and IPT therapists were not blind to group assignment.

## Statistical methods

**Univariate statistics.** All analyses were conducted using Stata 15 [46]. Using an intent-to-treat analysis, MDD and PTSD MINI scores were grouped as binary variables to indicate positive versus negative status at each assessment visit, with continuous measures for the BDI and PCL. At each assessment visit, BDI and PCL were analyzed as numerical scores. Additional baseline measures included: Alcohol Use (AUDIT), Drug Use (DAST), lifetime traumatic events (THQ), IPV in past week among partnered participants (CTS2), disability (WHODAS), days in the past month partially or completely unable to work, school fees paid on time in past month for those with school-aged children, and monthly income. We included baseline values in all analyses (as opposed to covariate) and used all relevant time points: baseline through end of follow for numeric, and month 3 through end of follow for binary outcomes that were 100% at baseline (MDD and PTSD) and were therefore fixed. We report the frequency of non-missing data out of the total number of participants in each group for each measure. To evaluate baseline between-group comparability, patient data were stratified by randomization group and tested for between-group differences. Similarly, each treatment group was stratified by participants who completed and did not complete the study and tested for baseline characteristic differences. The proportion of participants that did not complete the study at the 3-month assessment time point (IPT + TAU group) or at the 6-month assessment time point (TAU group) are reported, as well as the overall proportion of participants who did not complete each assessment. Continuous variables were described by means and compared by t tests and categorical variables were compared by chi-squared or Fisher exact tests. All tests were performed using an alpha level of 0.05.

**Multivariate statistics.** The primary outcomes (binary), MDD and PTSD, were separately analyzed using intention-to-treat methods with multilevel mixed-effects logistic regression to assess the differences between treatment groups over time. We included a random effect (intercept) for participant and fixed effects of treatment group, time (as a categorical variable), and the interaction between time and treatment group. Both groups started with all participants diagnosed with both MDD and PTSD on the baseline MINI v5 and so that time point was left out of the analyses. The main group comparison of interest was at week 12, when the IPT+TAU group had completed IPT and the Wait List+TAU group had not yet begun IPT.

Most of our secondary outcomes were analyzed using linear mixed models (with the same random and fixed effects as above), including BDI, PCL, and WHODAS. Physical IPV was derived from the CTS (above) and coded as a binary variable, because "no violence" is a goal of clinical care. Physical IPV was assessed in the subgroup of partnered participants using multilevel mixed-effects logistic regression to assess the interaction between treatment group for repeated measures over time. We included a random effect (intercept) for participant and fixed effects of treatment group, time (as a categorical variable), and the interaction between time and treatment group.

## Results

### Baseline univariate statistics

The great majority of participants self-referred after listening to a "health talk" at the clinic describing the study (90%), and the remainder were referred by HIV clinic staff and providers who also heard the health talk. Of the 382 women screened between May 2015 and July 2016, 126 (33%) were ineligible and 256 (67%) were eligible (Fig 1). At baseline, all 256 participants had positive MDD and PTSD diagnoses and were randomized to IPT and TAU ($n$ = 123) or TAU ($n$ = 133). Consistent with MINI v5 diagnoses, mean baseline MDD and PTSD symptom scores (BDI and PCL) were high across treatment groups. At baseline, over half of partnered participants reported experiencing physical violence from an intimate partner in the past week. Ninety percent reported at least 2 types of lifetime traumatic events, and over a third reported more than 4 different trauma types. At baseline, participants reported very high levels of disability secondary to health conditions. On average, they reported partial or complete health-related inability to work on 7 to 8 days over the past month. The average monthly income was about 18USD. Reported harmful alcohol or illicit drug use was low (Table 2). Only 1 of the IPT+TAU participants had ever received mental health care (lifetime), and only 5 of the Wait List-TAU group had ever received care. None of the participants were in current treatment for mental disorders.

**IPT therapist training outcomes.** IPT training began with 21 prospective IPT therapists. Ten of these joined the study after successfully completing the required didactic training and IPT practice cases (with average IPT adherence score of "9" or higher).

**Non-completer analyses.** Overall, 35% (90/256) of participants did not complete the trial, including its 4 assessment points. Among the 123 participants randomized to the IPT+TAU arm, 27.6% (34/123) did not complete IPT at the 3-month follow-up time point. Among the 133 randomized to the Wait List-TAU arm, 20.5% (24/117) did not complete the subsequent course of IPT sessions (Fig 1). At month 3, lost to follow-up (LTFU) was higher in the IPT +TAU group than in the Wait List+TAU group (27.6% [34/123] versus 12% [16/133], $p < 0.01$). When we stratified each treatment group by study completers versus non-completers, we found no significant differences in baseline depression (BDI) or PTSD (PCL) symptoms, nor in experience of past week IPV (see S2 Table for details).

At baseline, BDI depressive symptoms were high in both the IPT+TAU and the Wait List +TAU groups, with average scores of 27.5 and 29.0, respectively. The usual BDI cut score for

                    

likely clinical depression is 16; thus, the average group BDI scores correspond to moderate depressive symptoms and corroborate the diagnosis of MDD on the MINI. At baseline, both randomized groups had high mean PTSD symptoms on the PCL-C, the IPT+TAU group averaged 56.6 and the Wait List+TAU group 56.1, consistent with PTSD diagnosis (a validated cut score is 30).

**Effect of interpersonal psychotherapy on primary outcomes.** The percentage of IPT +TAU participants meeting MDD criteria dropped from 100% at baseline to 33.5%, 95% CI [22.9% to 44.0%] at month 3, while the Wait List+TAU group dropped from 100% to 57.6%, 95% CI [48.6% to 66.6%]—an MDD risk ratio of 0.64 for IPT+TAU participants. Across the 4 study time points, the only significant group for MDD was at month 3—the point at which IPT+TAU participants had completed treatment and Wait List+TAU participants had not yet begun. Participants who received IPT had lower odds of MDD (odds ratio (OR) after intervention 0.26, 95% CI [0.11 to 0.60], $p$ = 0.002) (Fig 2). The pattern of PTSD response was similar. At month 3, the percent of participants meeting PTSD criteria in the IPT+TAU group was 21%, 95% CI [12.2% to 30.6%], compared to 37%, 95% CI [28.4% to 46.1%] in the Wait List +TAU group—a PTSD risk ratio of 0.80 for IPT+TAU participants. Across the 4 study time points, the only significant group difference for PTSD diagnosis was at month 3. Participants who received IPT had lower odds of PTSD (OR after IPT 0.35, 95% CI [0.14 to 0.86], $p$ = 0.02) compared with controls (Fig 2). The pattern of comorbid MDD-PTSD was similar. At month 3, the percent of participants meeting MDD-PTSD criteria in the IPT+TAU group was 20%, 95% CI [10.8% to 28.7%], compared to 35%, 95% CI [26.0% to 43.3%], in the Wait List+TAU group. Across the 4 study time points, the only significant group difference for MDD-PTSD diagnosis was at month 3. Participants who received IPT had lower odds of MDD-PTSD (OR after IPT 0.36, 95% CI [0.15 to 0.90], $p$ = 0.03) compared with controls (Fig 2). Post-IPT gains were maintained at later time points without statistical differences.

There was a significant group difference for depression symptoms at month 3, when the 2 groups differed in their receipt of IPT. From baseline to 2 months, the IPT+TAU group average BDI scores dropped by over 10 points, whereas the Wait List+TAU group average BDI scores dropped 6 points (−4.95 [8.67, −1.22], $p$ = 0.009) (Fig 3). We used a cutoff of 16 for the BDI II for reference in Fig 2, which is supported for medically ill patients [47]. As with depression symptoms, there was a significant group difference for PTSD symptoms at 3 months, with the IPT+TAU group significantly decreased compared with the Wait List+TAU group which had not yet received IPT. From baseline to 3 months, the IPT+TAU group average PCL scores dropped over 17 points, whereas the Wait List+TAU group average PCL scores decreased over 9 points (−8.93 [−14.06, −3.81], $p$ = 0.001) (Fig 3). Post-IPT gains for both depression and PTSD symptoms were maintained at follow-up visits. We used a cut score of 30 for the PTSD symptoms on the PCL-C for references in Fig 2 [48].

**Secondary outcomes: Disability and absenteeism.** Across the 4 study time points, the only significant group difference for disability was at the month 3 (post IPT treatment) assessment. IPT+TAU group disability decreased 37.5% from baseline, whereas the Wait List+TAU group decreased by 15.2%. Participants who received IPT+TAU had 7% lower disability scores on the WHODAS 2.0 than those in Wait List+TAU (−6.9 [−12.2, −1.5], $p$ = 0.01). Gains were maintained across follow-up. There was also close to a significant effect for the number of days participants were partially or totally unable to work or carry out usual activities (absenteeism) over the past month. IPT+TAU recipients reported a 33% decrease in absenteeism after treatment (month 3), while Wait List+TAU participants had a 7% increase over that period. On average, IPT+TAU participants had a nonsignificant reduction of 3.4 days in past month absenteeism compared with Wait List+TAU participants at 3 months (−3.35 [−6.83, 0.14], $p$ = 0.06).

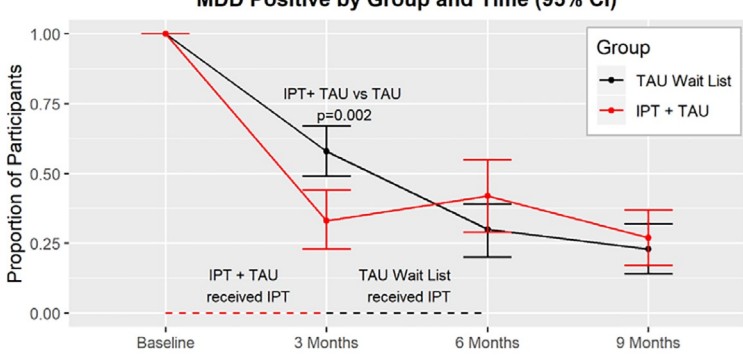

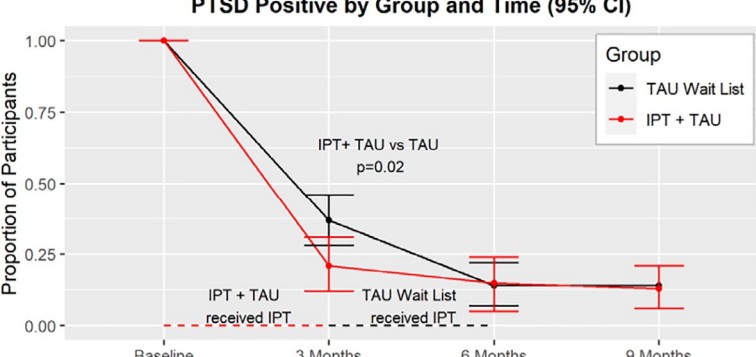

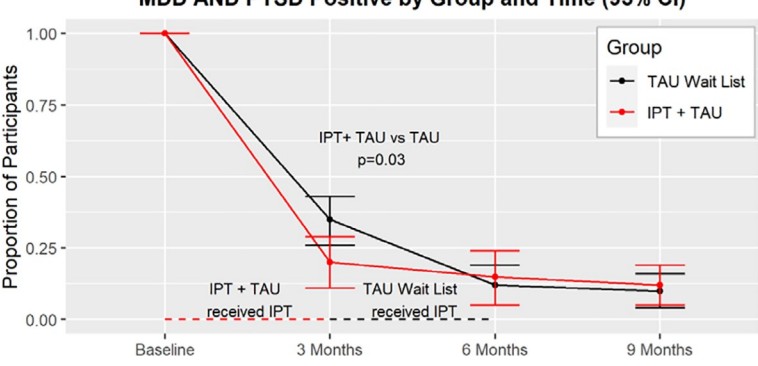

**Fig 2. Primary outcomes on the Mini International Neuropsychiatric Interview (MINI): Major Depressive Disorder and Posttraumatic Stress Disorder, *n* = 256.**

**Secondary outcomes: Violence outcomes for partnered participants.** While the study was not powered to detect change in IPV, we analyzed physical IPV among the smaller subset of participants who were partnered at each assessment point. Physical IPV in the IPT+TAU group decreased 71% from baseline, whereas physical IPV in the Wait List+TAU group decreased by 40%. A significant group difference was present for at the 3-month (post IPT treatment) assessment. In mixed effects ML regression modeling physical IPV score on time, we found −2.79 [−5.42, −0.16], *p* = 0.04. Gains were retained across follow-up (*n* = 27).

**Missing data analyses.** At the key time point (3 months), there were missing data on the primary outcomes for 1 completer in the Wait List+TAU group and 13 in the IPT+TAU group. We conducted a sensitivity analysis [49] by checking all possible allocations of the

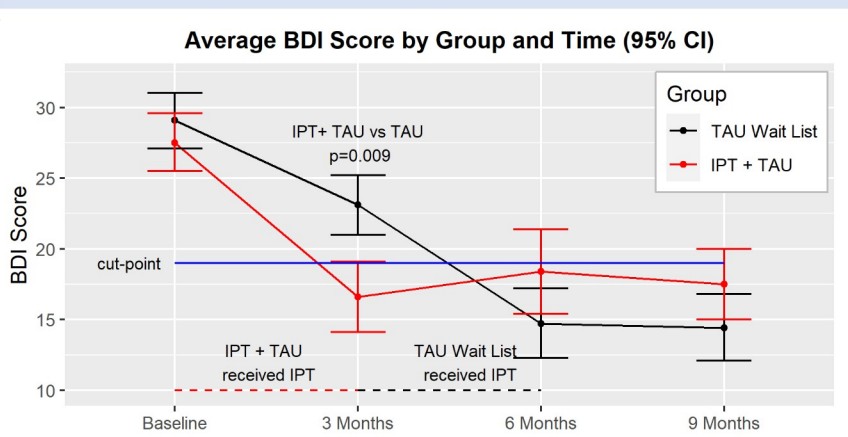

**Fig 3. Average depression and PTSD symptom scores.**

missing data (so 2*14 = 28 possibilities in all) to have or not have remission for that outcome. For each allocation, we recalculated the *p*-value. For the outcome of MDD, statistical significance was very robust to the missing data with only a single allocation not being statistically significant (allocating the missing Wait List+TAU to remission and none of the 13 in the IPT +TAU group to remission); even that allocation had a borderline *p*-value of 0.068. The analyses for PTSD and the combined MDD and PTSD outcomes were somewhat less robust. Using the observed rates, we would expect about a 50% chance of the Wait List+TAU missing data to be a remission and between 10 and 11 of the IPT+TAU group to achieve remission. If 8 or fewer of the IPT+TAU with missing data were allocated to be remitters, then statistical significance was lost.

## Implementation outcomes

**Feasibility.** We prespecified a feasibility benchmark of recruiting 220 participants by screening 300, or 36% more than enrolled. With the increased sample size that was used in the actual study (256), we screened 382 prospective participants, which is 50% more than enrolled —we had to screen more than we planned in order to obtain our target enrollment. Our next

feasibility benchmark stated that completers would attend 80% or more of their IPT sessions. This benchmark was achieved. We expected attrition to be 50 participants or more with a sample size of 220. Using the same ratio, we would expect attrition of 58 with our actual sample size of 256. Our attrition was 90, 55% higher than expected. Last, we included a feasibility measure in regards to therapist attrition—the study would be considered feasible if fewer than one-half of therapists left the study. This benchmark was met—no therapists left the study.

**Fidelity.** IPT adherence was scored using a 10-point Likert scale assessing adherence in each of the 3 IPT phases. A score of "0" indicated "not at all" adherent, and "10" indicated "fully adherent." Sessions were considered adherent to the IPT protocol if they averaged a score of 5 (moderate adherence) or higher and did not employ off-protocol interventions [24]. All the study IPT sessions scored higher than 5 on adherence assessments. Of note, while audio sessions were recorded, due to unexpected expenses, the budget did not allow for a randomized 20% of sessions to be independently evaluated as planned. However, we initiated local peer IPT supervision as an adjunct to and countercheck of adherence (see "Interventions" for details).

## Discussion

This study found IPT highly effective for comorbid MDD and PTSD among HIV–positive women affected by GBV, with high rates of remission, and clinically significant decrease of symptom severity for both disorders, with all improvements sustained over 6-month follow-up. Interestingly, the Wait List-TAU group decreased in MDD, PTSD, and corresponding symptoms between baseline and month 3, prior to receipt of IPT. This could be due to a number of reasons related to the study's setting and population: (1) for many of these women, this study was the first time they had ever been asked about their emotions and experiences with GBV by a sympathetic and respectful person—this, alone, could have been cathartic; and (2) as mentioned above, almost none of these women had ever received any formal mental health care—the hope that was instilled by their enrollment in the study and upcoming IPT could have improved their mental health even before treatment started. The MDD outcome was robust to missing data analyses, while the PTSD outcome was somewhat less so. In secondary outcome analyses, IPT decreased health-related disability and apparently improved employment outcomes for IPT+TAU participants relative to Wait List+TAU controls. IPT was also associated with decrease in physical IPV for the study's subset of partnered participants. Overall, the study showed that IPT effectiveness is strong in a "real-world" setting with delivery by local nonspecialists and integrated within existing HIV clinic operations.

In regard to implementation outcomes, IPT adherence was strong and local peer IPT supervision group was feasible. Despite fairly rigorous prespecified feasibility benchmarks, the study met 2 of the 4, missing the screening/enrollment window slightly and experiencing higher than expected attrition (see Limitations). As described, the study provided 300ksh (approximately 3USD) to participants to cover travel costs, given the distance they traveled for weekly IPT sessions. In our view, this is not a limitation of the study, but rather an important consideration for implementation of psychotherapy in LMICs. We look forward to calculating these transport costs as part of the IPT intervention with our forthcoming economic analyses.

The results contribute to our understanding of psychotherapy treatment outcomes among people living with HIV by showing that evidence-based IPT for depression and PTSD among HIV+GBV+ women can be feasibly integrated with HIV care and delivered in a patient-preferred setting with excellent mental health outcomes. Despite awareness of a high prevalence of depression among PLWH in East Africa, to our knowledge, this is one of fewer than 15 studies of depression treatment integrated with HIV care across SSA, the only study of integrated

treatment for comorbid depression and trauma-related disorders, and the only study of mental health treatment for the large, heavily affected group of HIV+GBV+ women in SSA. While this study used a hybrid effectiveness-implementation design type I weighted toward evaluation of effectiveness outcomes, we used scalable strategies, including integration with existing HIV care and use of a readily available workforce of nonspecialists. Given the extensive network of similar HIV clinics across East Africa, this study provides support disseminating this scalable delivery setting and workforce for treating MDD and PTSD among care-engaged HIV+GBV+ women. A notable exception is our IPT adherence monitoring system, which included telephone supervision and adherence scoring of IPT sessions by a psychiatrist and IPT expert in Kenya or US (LO and SMM). Investigators sought to address this issue by initiating on-site group supervision during the study (supplementing ongoing individual supervision). The system worked well, and local group supervision had primarily replaced the need for investigator supervision of IPT by the time of study conclusion.

In the setting of HIV clinical care, prior research shows that depression and PTSD have strong associations with decreased ART adherence and HIV treatment attrition [5,6]. While the women in our study were engaged in HIV care, such research suggests that they faced elevated risk of ART default or attrition from care because of their psychiatric diagnoses. In this era of aggressive progress toward HIV epidemic control, we cannot afford to miss high-risk cases by failing to apply evidence-based, cost-effective solutions, such as the intervention in this study.

## Scaling East African mental healthcare for HIV–positive women affected by gender-based violence

The HIV–positive women in this study were engaged in care at one of many SSA HIV clinics supported by The President's Emergency Plan for AIDS Relief (PEPFAR). In 2019, PEPFAR supported HIV care for an estimated 9.3 million women and girls. This is not only a huge number of treatment-engaged individuals, but an easy starting point in the delivery of efficient, effective mental healthcare for HIV–positive women with high GBV prevalence and associated depression and trauma disorders. HIV+GBV+ women engaged in ongoing care demonstrate courage and motivation by seeking HIV treatment for a heavily stigmatized disease, and they return regularly to the clinic when they are typically in a non-acute state of health. For HIV+GBV+ women with depression or trauma-related disorders, these characteristics make them ideal candidates for successful mental health treatment. Even if scaling up treatment for depression and PTSD treatment for HIV+ women affected by GBV in SSA were limited to those already engaged in HIV care—the easiest group to access—the public health impact would be substantial.

## Limitations

This study had a relatively high rate of attrition. Approximately 29% of those who left the study cited relocation or transfer of HIV care as the reason for dropout. This might reflect the documented high prevalence of migration among HIV–positive women in SSA [50]. Another 27% of those who left the study reportedly could not attend sessions due to time constraints, commute, or work conflict. The large catchment area of the health center suggests that weekly IPT could have been an overly burdensome time commitment, particularly for women who lived far from the health center. While the primary outcomes and GBV exposure did not differ between completers and non-completers at baseline, it is possible that some participants responded more quickly than others, leading to premature study discontinuation among healthier participants, as their disability lifted and they pursued work opportunities, which

often involve local migration. Attrition of healthier participants in the IPT group would bias toward the null hypothesis. There was greater attrition from the IPT+TAU group than the Wait List+TAU group between baseline and 3 months. This could reflect a degree of unacceptability of IPT for some participants.

We intentionally restricted our study sample to women who were already engaged in HIV care. As such individuals are more likely to be motivated for treatment, IPT uptake may have been higher in our study than it would be for patients not yet engaged in HIV care. However, considering the many women with MDD and PTSD currently enrolled in HIV care, beginning scale up with this treatment-engaged group in urgent need of mental healthcare is a logical starting point that could prevent attrition from care and associated resources required to reengage.

The meaning of TAU differs across studies. Here it refers to the usual psychosocial resources available through the FACES clinic, as described above. The study did not collect data on which TAU resources were accessed by the participants and therefore could not adjust for their effect. Given that TAU was available to all participants in both groups throughout the study and the randomized design, it is unlikely to have a significant effect on study results for IPT effect.

The study was not able to determine why participants who received IPT+TAU were less likely to be in a violent intimate partnership, compared with controls. We note that the percentage of study participants who reported having intimate partners decreased from 43% at baseline to 16% at the final, 9-month assessment. If those in violent relationships were more likely to leave partners or leave the study, this could bias results toward decreased violence among those who completed the study. In addition, study participants experiencing physical IPV might have been more likely to leave IPT+TAU, which could account for the apparent effect of IPT on violence reduction. However, if study participants experiencing physical IPV were more likely to leave the study, then, given (1) the randomized design; (2) comparable baseline partner violence between IPT+TAU and Wait List-TAU (Table 2); and (3) comparable baseline partner violence between IPT+TAU completers and non-completers (appendix), we would not expect to find a significant effect of IPT+TAU on violence at 3 months. If study participants experiencing physical IPV were more likely to leave IPT+TAU (versus Wait List-TAU), we would not expect our finding of continued reduction of violence after IPT treatments (e.g., between months 6 and 9).

## Conclusions

This study in western Kenya shows that local nonspecialists with high school–level education can be trained to successfully deliver IPT integrated with HIV clinic operations for MDD and PTSD among HIV+ women affected by GBV, with sustained benefit for recipients. While our model does not require prior mental health training for IPT providers, IPT training and practice with multiple levels of ongoing supervision did support the intervention. This type of training collaboration between mental health professionals and nonspecialists with cascading levels of supervision appears to be a promising strategy for scale up of care that might enlist national mental health professionals in leadership for scale up.

## Supporting information

**S1 Table. CONSORT 2010 checklist of information to include when reporting a randomized trial.**
(DOC)

**S2 Table. Baseline characteristics of completers and non-completers.**
(DOCX)

**S1 Text. Excerpt from Mental Health and Domestic Violence Crisis Protocols: MIND Study.**
(DOCX)

## Author Contributions

**Conceptualization:** Susan M. Meffert, Thomas C. Neylan, Elizabeth A. Bukusi, Helen Verdeli, Harsha Thirumurthy.

**Data curation:** Susan M. Meffert, Charles E. McCulloch, Kelly Blum, Grace Rota, Ray Rota, Elizabeth Opiyo.

**Formal analysis:** Susan M. Meffert, Thomas C. Neylan, Charles E. McCulloch, Kelly Blum, James G. Kahn, Harsha Thirumurthy.

**Funding acquisition:** Susan M. Meffert, Thomas C. Neylan, Craig R. Cohen, Elizabeth A. Bukusi.

**Investigation:** Susan M. Meffert, James G. Kahn, Linnet Ongeri.

**Methodology:** Susan M. Meffert, Thomas C. Neylan, Helen Verdeli.

**Project administration:** Susan M. Meffert, Craig R. Cohen, Grace Rota, Ray Rota, Grace Oketch, Elizabeth Opiyo, Linnet Ongeri.

**Resources:** Susan M. Meffert, Craig R. Cohen.

**Software:** Susan M. Meffert, Charles E. McCulloch.

**Supervision:** Susan M. Meffert, Thomas C. Neylan, Charles E. McCulloch, Elizabeth A. Bukusi, Helen Verdeli, James G. Kahn, Linnet Ongeri.

**Validation:** Susan M. Meffert.

**Visualization:** Susan M. Meffert.

**Writing – original draft:** Susan M. Meffert.

**Writing – review & editing:** Susan M. Meffert, Thomas C. Neylan, Charles E. McCulloch, Kelly Blum, Craig R. Cohen, Elizabeth A. Bukusi, Helen Verdeli, John C. Markowitz, David Bukusi, Harsha Thirumurthy, Grace Rota, Ray Rota, Grace Oketch, Elizabeth Opiyo, Linnet Ongeri.

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
