## [Editor Report · Decision Letter 0]

19 May 2020

Dear Dr Meffert, 

Thank you for submitting your manuscript entitled "Effectiveness of Interpersonal Psychotherapy Delivered by Non-Specialists for Depression and Posttraumatic Stress Disorder Among Kenyan HIV-Positive Women Affected by Gender-based Violence" for consideration by PLOS Medicine.

Your manuscript has now been evaluated by the PLOS Medicine editorial staff and I am writing to let you know that we would like to send your submission out for external assessment.

Kind regards,

Richard Turner, PhD

Senior editor, PLOS Medicine

rturner@plos.org

---

## [Decision Letter · Decision Letter 1]

25 Aug 2020

Dear Dr. Meffert,

Thank you very much for submitting your manuscript "Effectiveness of Interpersonal Psychotherapy Delivered by Non-Specialists for Depression and Post-traumatic Stress Disorder Among Kenyan HIV-Positive Women Affected by Gender-based Violence" (PMEDICINE-D-20-02128R1) for consideration at PLOS Medicine. 

Your paper was evaluated by an academic editor with relevant expertise, and sent to independent reviewers, including a statistical reviewer. The reviews are appended at the bottom of this email and any accompanying reviewer attachments can be seen via the link below:

[LINK]

In light of these reviews, we will not be able to accept the manuscript for publication in the journal in its current form, but we would like to invite you to submit a revised version that addresses the reviewers' and editors' comments fully. You will appreciate that we cannot make a decision about publication until we have seen the revised manuscript and your response, and we expect to seek re-review by one or more of the reviewers. 

We hope to receive your revised manuscript by Sep 15 2020 11:59PM. Please email us (plosmedicine@plos.org) if you have any questions or concerns.

Please let me know if you have any questions. Otherwise, we look forward to receiving your revised manuscript in due course. 

Sincerely,

Richard Turner, PhD

rturner@plos.org

To your title, please add a study descriptor following a colon, e.g., "...: a randomized controlled trial".

Please combine the "Methods" and "findings" section of your abstract, in accordance with journal style, and add a new final sentence to the new combined subsection quoting 2-3 of the study's main limitations. 

After the abstract, we will need to ask you to add a new and accessible "author summary" section in non-identical prose. You may find it helpful to consult one or two recent research papers published in PLOS Medicine to get a sense of the preferred style. 

To your methods section, please add some additional detail on study ethics approvals, e.g., date of approval and application numbers. Please state whether consent was written or verbal, and whether informed. 

Please remove the results of baseline statistical tests from table 1.

Please remove the statement about data sharing from the end of the main text - this information will appear in the article metadata in the event of publication, via entries in the submission form. Please also consult our policy on data sharing - a non-author contact for inquiries about access to study data will be needed (https://journals.plos.org/plosmedicine/s/data-availability). 

Please make that "sub-Saharan Africa" throughout. 

Please state exact p values or "p<0.001".

Throughout the text, please adapt reference call-outs to the following style: "... general population [1,2]." (i.e., no spaces within the square brackets). 

In your reference list, please convert italics into plain text. Where appropriate, 6 author names should be listed rather than 3, followed by "et al.".

Please add a completed CONSORT checklist as a supplementary document, referred to in your methods section (e.g., "See S1_CONSORT_Checklist"). In the checklist, individual items should be referred to by section (e.g., "Methods") and paragraph number rather than by line or page numbers, as the latter generally change in the event of publication. 

Comments from the reviewers:

*** Reviewer #1: 

This review relates to manuscript PMEDICINE-D-20-02128 about Effectiveness of Interpersonal Psychotherapy Delivered by Non-Specialists for Depression and Posttraumatic Stress Disorder Among Kenyan HIV-Positive Women Affected by Gender-based Violence.

I have listed my comments, major or minor, below.

* Major comments:

o I believe the flow of the manuscript could be improved. In particular, a lot of information included in the "Study design" section consists in background information. These two sections need to be reorganised. 

o I am concerned about potential multiplicity issues due to the number of primary outcomes. It is also unclear which outcomes should be considered primary. The outcome section list 4 'primary outcomes'. The authors also write that they used both continuous and dichotomous versions of MDD and PTSD. In other places only MDD and PTSD are listed as the two primary outcomes. Please clarify as well as discuss the potential multiplicity issue including how one might control the risk of false-positive results in light of the number of tests conducted.

o Randomization: A colored Excel list was used as the randomisation sequence. If the study coordinator performing the randomisation has access to the Excel file, s/he was able to identify upcoming allocations. This lack of concealment comes with a potential high-risk of bias. Please clarify whether future allocations were concealed and if measures were in place to ensure no preferential allocation (e.g. skipping an allocation to 'force' a participant to be allocated to a specific group).

o Given the substantial amount of participants with missing outcome data, I would suggest some sensitivity analyses to assess the robustness of the findings under various assumptions about the missing data. For example, one could use a range of simple imputations to determine the tipping point, or the point at which the results are no longer consistent with the main (non-imputed) findings. Please refer to the following paper for example: https://pubmed.ncbi.nlm.nih.gov/12483769/

o I would strongly encourage all pre-specified study outcomes to be reported in this main manuscript. I would expect to see at least all primary and secondary outcomes as well as details of the intervention delivery and adherence. I believe a decision not to report all outcomes should be reported by strong justification.

o Please attach the study protocol and/or statistical analysis plan as a supplement. The statistical analysis plan should clearly indicate which analyses were pre-specified vs post-hoc, i.e. decided after seeing unblinded results

* Minor comments:

o Abstract: please indicate the specific setting (e.g. city + country)

o Abstract: please indicate the timing corresponding to the primary outcome results e.g. 3 months

o Abstract: please correct typo in confidence interval for OR (0.0.60)

o Objective section: please clarify what a "Type I trial" is. 

o Objective section: The authors write that "The diversity of secondary study outcomes exceeds the scope of a single manuscript". It may indeed be difficult to report all outcomes in a single manuscript; however, I would like to see a complete list of all pre-specified study outcomes listed accompanied by a clear indication of which ones are reported in the current manuscript. According to the ClinicalTrial.gov record, there is only a limited number of outcomes measures in which case I believe all should be reported in the current paper.

o Please consider adding details of the intervention including its various components. This could be included as a supplement.

o Methods: consider moving the sub-section labelled "HIV clinic setting" to the background section.

o Methods, study design: the first paragraph repeats the objective sub-section. Please remove the repeated information from the objective sub-section and consolidate. I would suggest that the second paragraph (starting with "Wait List-TAU comparators for psychotherapy…") be moved to the background section.

o Sample size: I note that the sample size was calculated to detect an effect size (Cohen's d) of 0.8 which is very large. Apart from the earlier study (reference 41) used as a reference, are there other sources supporting such a large effect?

o Multivariate statistics: please add details about the main model. In particular, please clarify the multilevel components including what were the level(s) and how the structure (correlations) were accounted for. Please explicitly list all fixed and random effects included in the model. Was the baseline value of the outcome (MDD or PTSD) included as a covariate?

o Please indicate the statistical software(s) used for the analyses

o Please include a table comparing baseline characteristics of completers vs non-completers in the online supplement.

o When reporting the effect of the intervention on outcomes, the authors talk about significant group by time interaction at specific timepoints. For example: "the only significant group by time interaction for MDD was at month three". In this case, it is not the interaction that is significant but the difference between intervention arms at a specific timepoint. The interaction measures the overall consistency of the intervention effect across the 3 timepoints. Please amend accordingly.

o Please describe adherence to the intervention. For example, duration of the sessions, proportions attending each session, etc.

o When reporting secondary outcomes (disability and absenteeism), the authors mention "controlling for the effect of time..". Please clarify what this means. Is this using a different model than for the primary outcome(s)?

- Laurent Billot

*** Reviewer #2: 

Dear Authors

Thank you for the opportunity to review this manuscript reporting individually randomized 2-arm trial of an interpersonal psychotherapy for HIV-GBV+ women with co-morbid depression and PTSD in a low resource setting. Areas for improvement include explicitly reporting data as required by the CONSORT statement. For example, explicitly reporting the randomization ratio. I suggest the authors to revise the manuscript against the CONSORT checklist, add missing information and submit revised manuscript alongside completed CONSORT checklist. 

I have never come across "a hybrid effectiveness-implementation design type I weighted toward evaluation of effectiveness outcomes". To my view, this study is a standard individually randomized 2-arm trial. If the authors insist on the hybrid design, this choice should be justified and supported by relevant references.

The follow up period of 6 months for the intervention arm and 3 months for the control arm is not enough to prove the sustainability of the intervention effect. Authors' claims about sustainable effect should be toned down. 

p. 13. It seems that some text slipped during formatting. Paragraph under Table 1 should be under Baseline Univariate Statistics not under Non-Completer Analyses.

*** Reviewer #3: 

This study is an important contribution to the literature as it describes the results of a fully-powered RCT effectiveness-implementation hybrid design to test the effectiveness of IPT in addressing comorbid MDD and PTSD in GBV-affected HIV-positive women in Kenya, which is a highly understudied population. Moreover, the providers were non-specialists and treatment was delivered in HIV clinics, which makes it a novel and potentially more sustainable and scale-able intervention. 

Overall the manuscript is quite strong. I have listed comments below that may further strengthen this article.

The authors chose to focus exclusively on improvement of mental health outcomes in the manuscript, but the study is framed specifically as an "effectiveness-implementation" trial. It would have been helpful to learn more about the implementation outcomes of the trial, in addition to changes in the mental health outcomes. For example, information on feasibility, fidelity, acceptability, satisfaction, and barriers and facilitators to treatment delivery would have been useful in understanding and interpreting the outcomes.

Similarly, although the intervention specifically targets HIV+ women who have experienced GBV, changes in GBV and HIV adherence/viral load were not reported. The authors note that they have space limitations, but it would be helpful for the reader to have these outcomes reported.

PARTICIPANTS: The authors note that participants were diagnosed using the MINI 5.0. Who conducted the assessments? What was their training? Were women currently experiencing GBV, and if so, how was risk of harm addressed during the study? Additionally, how were exclusion criteria of severe thought and mood disorder and current drug and alcohol dependence assessed and addressed? By whom? Did the participants receive any compensation for assessments, treatment visits, or transportation costs?

RECRUITMENT. The authors state that "Patients were recruited through FACES HIV provider referral and self-referral following informational talks to FACES patients in the HIV clinic check-in and waiting areas. Consecutive HIV-positive women were consented and screened for inclusion from May 2015-July 2016." How many women were referred by providers out of the total case load? What criteria did providers use to determine whether referral was appropriate? How many women were given the informational talks? How many self-referred? Of the provider vs. self-referrals, how many met inclusion and exclusion criteria?

How many prospective IPT therapists applied and how many went on to meet the requirements of being a therapist?

RESULTS. Participants in the intervention group had significantly greater symptom and functional improvement compared to the TAU group. However, the TAU group also showed improvements on depression, PTSD, and functional impairment outcomes. The improvement is so marked that participants in TAU had PCL scores that were below the clinical cut-point before they received the intervention. This improvement in symptom change is particularly notable given that all participants met MINI criteria for MDD and PTSD at baseline. What do you think may be accounting for such improvement in the TAU condition?

SMALL Typos:

"At each assessment visit, BID and PCL were analyzed as numerical scores". Should be BDI

"At month three, LFTU was higher in the IPT+TAU group than the Wait List+TAU (27.6% (34/123) vs. 12% (16/133), p<0.01)." LFTU is not defined.

***

[LINK]

---

## [Decision Letter · Decision Letter 2]

28 Oct 2020

Dear Dr. Meffert,

Thank you very much for re-submitting your manuscript "Interpersonal Psychotherapy Delivered by Non-Specialists for Depression and Posttraumatic Stress Disorder Among Kenyan HIV-Positive Women Affected by Gender-based Violence: Randomized Controlled Trial" (PMEDICINE-D-20-02128R2) for consideration at PLOS Medicine.

I have discussed the paper with our academic editor and it was also seen again by 3 reviewers. I am pleased to tell you that, provided the remaining editorial and production issues are dealt with, we expect to be able to accept the paper for publication in the journal.

The remaining issues that need to be addressed are listed at the end of this email. Any accompanying reviewer attachments can be seen via the link below and the attached reports. Please take these into account before resubmitting your manuscript:

[LINK]

Please let me know if you have any questions. Otherwise, we look forward to receiving the revised manuscript shortly. 

Kind regards,

Richard Turner, PhD

rturner@plos.org

Requests from Editors:

Please revisit your data statements, which seem inconsistent - likely the answer to the first query should be "No - there are some restrictions". Please substitute a non-author contact for readers interested in inquiring about access to study data, so as to comply with our data policy (https://journals.plos.org/plosmedicine/s/data-availability).

Please add a "Secondary outcomes included ..." sentence early in the "Methods and findings" subsection of your abstract.

Please quote aggregate demographic details for study participants in your abstract.

We ask you to revisit the presentation of your findings in the abstract. For example, to improve clarity "... had 74% lower odds of MDD (0.256, 95% CI [0.11 to 0.60], p=0.0020 ..." should be amended to "... had lower odds of MDD (odds ratio [OR] 0.256, 95% CI 0.11 to 0.60, p=0.002) and PTSD (0.35, 0.14 to 0.86, p=0.02) ...". 

As you know, the risk of type I error is likely to be increased in the determination of secondary outcome findings. Therefore, please use an element of caution in presenting these findings, e.g., "...IPT+TAU participants had an apparent reduction in disability ...". Please introduce similarly softened language into the author summary.

We suggest adding an additional sentence, say, to the abstract to note the post 3 months follow-up points, and we imagine that you might wish to note the lack of difference in primary outcome measures post 3 months, as expected from the cross-over in provision of IPT.

Please modify the relevant sentence of your abstract to begin "Study limitations included ..." and remove the numbering of individual points. 

Please ensure that findings are quoted consistently throughout the ms (e.g., abstract "...lower odds of MDD [0.256 ...]"; results section of main text "0.26").

Please trim the "do and find" subsection of your author summary, limiting it to 3-4 short points. Many of the numbers here are also quoted in the abstract, and we suggest describing the findings in a narrative fashion in the author summary to avoid duplication. 

In the abstract, author summary and any other relevant points in the paper, please do not describe non-significant findings with the word "trend". 

You mention "space limitations" in your methods section. Technically, PLOS Medicine does not have space limitations (considering the option of supplementary information) and so we suggest removing this wording.

Assuming this is the case, please note in the methods section (main text) that informed consent was sought, and by what means. 

At the start of the results section (main text), please avoid "approximately 90%" and "roughly 10%". Under "Effect of Interpersonal Psychotherapy ...", please substitute an exact number for "roughly 37% (95% CI 28.4 to 46.1%)" and similar quantities. 

On p.35 of the PDF, please correct "... was a significant group difference ... was at month 3 ...".

Again in the results section of your main text, please remove the element of repetition in statements like "74% lower odds of MDD (OR 0.26 ...)", where "74% can be removed. 

In Fig 2, middle panel, "p=0.02" is quoted for "IPT+TAU vs TAU", yet what are presumably 95% CI appear to overlap. Why is this? A similar observation would seem to apply to the bottom panel. Incidentally, brief legends for figures 2 and 3 might be helpful to readers. 

Please refer to the attached CONSORT checklist at an appropriate point in your methods section (e.g., "See S1_CONSORT_Checklist"). 

Throughout the text, please adapt reference call-outs to precede punctuation consistently. 

Please remove periods from all subsection headings. 

Please remove all iterations of "[Internet]" from the reference list.

Noting reference 5, please ensure that all references have full access information. 

Noting the appendix table, we understand that CONSORT discourages the presentation of statistical tests of baseline quantities. 

Please remove page numbers from your attached CONSORT checklist. Individual items should be referred to by section (e.g., "Methods") and paragraph number rather than by page or line numbers, as the latter generally change in the event of publication. 

Comments from Reviewers:

*** Reviewer #1: 

My comments have been adequately addressed by the authors and I only have a few minor comments listed below:

In the section describing the models, please indicate which visits were included (e.g. Month 3 only or Months 3, 6 and 9), indicating that the value collected at the baseline visit was included as part of the outcomes (rather than as a covariate).

In the new supplemental table comparing completers vs non-completers, please consider combining the two randomised arms and compare all completers (both arms combined) to all non-completers (both arms combined). Given the relatively small sample size, I would also suggest not to rely solely on p-values to signal potential baseline imbalances.

In the discussion, please consider including a comment about the results of the sensitivity analyses conducted to explore the potential impact of missing data.

-Laurent Billot

*** Reviewer #2: 

Dear authors, I revised your responses to my comments. This is to confirm that you adequately addressed all the comments I made. 

*** Reviewer #3: 

The authors have addressed my comments.

***

[LINK]

---

## [Editor Report · Decision Letter 3]

3 Dec 2020

Dear Dr. Meffert, 

On behalf of my colleagues and the academic editor, Dr. Vikram Patel, I am delighted to inform you that your manuscript entitled "Interpersonal Psychotherapy Delivered by Non-Specialists for Depression and Posttraumatic Stress Disorder Among Kenyan HIV-Positive Women Affected by Gender-based Violence: Randomized Controlled Trial" (PMEDICINE-D-20-02128R3) has been accepted for publication in PLOS Medicine. 

PRODUCTION PROCESS

Before publication you will see the copyedited word document (within 5 business days) and a PDF proof shortly after that. The copyeditor will be in touch shortly before sending you the copyedited Word document. We will make some revisions at copyediting stage to conform to our general style, and for clarification. When you receive this version you should check and revise it very carefully, including figures, tables, references, and supporting information, because corrections at the next stage (proofs) will be strictly limited to (1) errors in author names or affiliations, (2) errors of scientific fact that would cause misunderstandings to readers, and (3) printer's (introduced) errors. Please return the copyedited file within 2 business days in order to ensure timely delivery of the PDF proof. 

If you are likely to be away when either this document or the proof is sent, please ensure we have contact information of a second person, as we will need you to respond quickly at each point. Given the disruptions resulting from the ongoing COVID-19 pandemic, there may be delays in the production process. We apologise in advance for any inconvenience caused and will do our best to minimize impact as far as possible.

EARLY VERSION

PRESS

PROFILE INFORMATION

Thank you again for submitting the manuscript to PLOS Medicine. We look forward to publishing it. 

Best wishes, 

Richard Turner, PhD

Senior Editor 

PLOS Medicine

plosmedicine.org